# Development and Validation of a Faith Scale for Young Children

Sungwon Kim

Department of Education, Chongshin University, Seoul 06988, Korea; swkim@chongshin.ac.kr

**Abstract:** The purpose of this study is to develop and validate a faith scale for young children. Data were collected from 424 young children, who had not yet entered elementary school, with their parents rating their faith level. Sixty-five preliminary questions were formulated under three domains—knowing, loving, and living—that were based on existing studies related to faith. The questions were reduced to 40 through a content validity test conducted by a seven-member panel. These questions were subsequently refined through pilot study, main survey, and statistical analysis. After exploratory and confirmatory factor analysis, the scale was finalized, comprising 25 questions that can be categorized into three factors: *confessional faith life*, *missional life*, and *distinctive life*. This scale is expected to measure early childhood faith and prove the effectiveness of Christian education programs on a young child's faith development.

**Keywords:** young children; faith scale; scale development; validation





## 1. Introduction

Psychologists have been interested in human religion and spiritual experience for more than a century (Moradshahi et al. 2017). Over the last 30 years, nearly 300 quantitative studies have been conducted on adolescents and adults' spirituality and well-being, while only a small number of studies about children, mainly using qualitative methods, have been conducted (Fisher 2015).

This researcher reviewed existing Korean studies related to this research topic by using the keywords "young children and faith" and "young children and spirituality" in the search engine KISS (Koreanstudies Information Service System) and RISS (Research Information Sharing Service). Among the studies published over the past 11 years (2010–2020) as the academic journal are 14 literature reviews with more than 80% as theory-based (Cho 2016b; Choi and Han 2011; Heo 2017; Joo 2018; Kang 2010; Kim 2010, 2012, 2015; Kim and Heo 2017; Kweon et al. 2018; Lee 2017; Park 2010; Seo 2013; Shin 2011); 10 Christian program development or experimental studies on the effectiveness of the program (Chung and Jeong 2014; Go 2010; Jeoung and Jin 2017; Lee et al. 2013; Lee and Kim 2010; Lee and Lee 2016; Ma and Kim 2020; Park 2014; Son et al. 2010; Song 2011); one investigative study on variables influencing young children's faith (Jung 2010), two case studies (Jung and Jeoung 2015; Yoo and Jeoung 2015), and three qualitative studies (Kim 2020a, 2020b; Min and Kim 2011). More specifically, research on program development has shown limitations, as many studies have examined dependent variables that are unrelated to faith (Cho 2012: depression and self-resilience; Jeoung and Jin 2017: character; Kim 2011a: aggressive behavior and self-control; Kim 2011b: social behavior and self-control; Ryu 2011: aggressiveness; Son et al. 2010: self-efficacy) or do not demonstrate its effectiveness. Additionally, it is regrettable that existing research on the scale development with theological problems (Kwon 2013), the use of subjective judgment by interviewers (Go 2008; Song 2005), the use of foreign tools, including theological doubts (Lee 2007), and the use of adult tools through translation and modification (Kim 2014) are doctoral dissertations. Nonetheless, these scales are used repeatedly in various journal articles (Chung and Jeong 2014; Lee 2011; Lee and Kim 2010; Lee and Lee 2016; Oh and Jung 2009).

Spirituality or faith-related research is focused on age groups older than young children, and research methods are concentrated in literature review (domestic) and qualitative study (foreign). One of the reasons for this phenomenon may be that there is no proper scale to measure the faith of young children or that there is a problem with it. Despite the recent rapid spread of spirituality scales, most of these scales are not easy to develop and have been developed for adults (Moore et al. 2016). Given such circumstances, it is difficult to find age-appropriate scales for children that can measure their spiritual sensitivity (Stoyles et al. 2012).

Existing studies related to the development of child faith/spiritual scales are as follows: Song (2005) developed an interview tool based on the interview guidelines of Lee (1993) paper, referring to interview questions from Fowler (1981) and Goldman (1984) to identify young children's faith development. It consists of 28 questions regarding five factors: *image of God* (8 questions), *faith and resurrection in Jesus* (7 questions), *heaven* (4 questions), *prayer* (4 questions), *forgiveness* (5 questions). Using Elkind (1971) analysis of children's religious concepts and Goldman (1965) study, the scoring method is given zero point for cases that do not respond or do not know the answer, one point for preoperational religious thought, and two points for concrete operational religious thought. For example, regarding the question, "What did Jesus do?" zero points for non-response or not knowing, one point for helping us, and two points for miracles are given. The tool shows problems in the logic of the questions, the theological soundness of the sample answers, and the subjective criteria for grading.

Kim (2014) study used a reconstruction of Oh and Kang (2000) and Ko (2006) scale, which originated from Howden (1992) *Spirituality Assessment Scale*, to identify the spirituality of young children. The factors of the spirituality scale consist of *energy in life*, *relationships with others and objects*, *integrated energy*, and *a positive perspective on the future*. The fact that Howden (1992) study was for adults aged 40–60, and there is no question about the relationship with God can be a problem. There are 20 questions including the word "friend" in this spirituality scale. For example, "be popular among friends," (no. 13) "try to hold it in if my friend gets annoyed," (no. 33) and "act as a mediator when my friends fight." (no. 39) In addition, doubts remain about the composition of factors given that the *positive perspective on the future* factor includes helping friends, finding meaning in teachers' praise, and saving water and electricity.

Stoyles et al. (2012) developed *The Spiritual Sensitivity Scale for Children (SSSC)* in a study of 7- to 14-year-old children. Its sub-factors are *outward focus* and *inward reflective focus*. While this scale reflects the relationship between oneself, others, and the world, it does not include the question of relationship with God (Moore et al. 2016). The following examples illustrate this point: "When I am really concentrating on something, I do not notice other things around me." (no. 3); "I am always learning new things about other people, the world, and myself." (no. 6); "I want to help others." (no. 12); "It is important to make sure my friends and family know that I love them." (no. 21)

The *Feeling Good Living Life* scale, developed by Fisher (2004), consists of 16 questions that correspond to *self-concern*, *family*, *environment*, and *relationship with God*. Each question is asked twice for two domains, "feeling good" and "living life" using a five-point Likert scale. Examples of questions are as follows: "Do you love your family?" (no. 1); "Do you know your family love you?" (no. 2); "Do you know your god is a friend?" (no. 6); "Do you want to be in the garden?" (no. 9); "Do you feel happy?" (no. 13); "Do you hear people say you are good?" (no. 14). In the Lee and Lee's study (Lee and Lee 2016), Fisher (2004) *Feeling Good and Living Life* scale were translated and modified to suit the Korean situation to measure the spiritual well-being of young children.

Moore et al. (2016) developed the *Children's Spirituality Measure* to identify the spiritual characteristics of children commonly applied to various religious traditions. The scale consists of a *comfort* factor, including 15 questions, *an omnipresence* factor containing eight questions, and *a duality* factor with four questions. Examples of questions are as follows: "I pray to G-d or talk to G-d when I feel sad." (no. 1; *comfort* factor); "G-d always knows

how I feel, even without talking." (no. 16; *omnipresence* factor); "Every person has a body and something inside them, like a soul or spirit." (no. 24; *duality* factor) Despite being a scale with a religious pluralistic background, the fact that it consists only of questions about God is unique. However, it is unfortunate that only three domains among God's various attributes and works are measured. It is likewise questionable whether dualism is meaningful as a subfactor for the scale for children.

Tratner et al. developed a *Childhood Religious Experience Inventory* for parents (Tratner et al. 2017a) and friends (Tratner et al. 2017b). The inventory for parents consists of four factors, namely, *assurance*, *disapproval and punishment*, *social involvement*, and *encouraged skepticism*. By looking only at parents' use, this scale allowed adult children aged 18–34 to evaluate the contents of their parents' religious guidance while recalling their childhood. It consists of 16 questions, four for each factor. Examples of questions include "Primary caregiver told me that God has blessed our family." (no. 2) and "Primary caregiver signed me up for a religious activity." (no. 9) The limitations of this inventory are threefold: it measured the influence of parents, not the child's own faith; it was used as a way to recall grown-ups' parental influence as a child; it omitted confirmatory factor analysis and merely conducted exploratory factor analysis. Information related to children's faith or spiritual scales is summarized in Table 1. Although the meaning of spirituality varies depending on the individual and religions, the universal meaning of spirituality is the process of pursuing the ultimate value an individual believes in and creating a mature life by integrating this value in the real life (You 2009). Citing Toon (1989); Anthony (2006) defines spirituality as "all about the human search for identity, meaning, purpose, God, self-transcendence, mystical experience, integration and inner harmony." (p. 6) Yeon (2004) defined spirituality as an insight into the transcendent aspects of human beings and their relationships with all worlds. Therefore, spirituality can be associated with God, nature, others, and the surroundings (Paul-Victor and Treschuk 2020). According to Lunn (2009) who distinguished between spirituality and faith, spirituality is an individual's beliefs and experiences in the supernatural realm, while faith is the belief or trust in a transcendent reality. Faith is an act of spiritual knowledge that involves beliefs with strong intellectual tendencies, formed by influential spiritual mediators or indirect religious and cultural factors (Kim 2006). According to Paul-Victor and Treschuk (2020), faith is often associated with spirituality and religion, but it is more personal, subjective, and deeper than these. As such, some scholars distinguish between spirituality and faith and some have a preferred term. However, this study views faith and spirituality as similar and compatible terms based on Christian truth and tradition.

Embracing these trends in previous research and the limitations of developed scales, the purpose of this study is to develop a faith scale for young children that is theologically sound, faithful to the development theories, and scientifically proven. The results of this study provide scholars and field practitioners with what constitutes the faith of young children and what to teach in early childhood. This work provides the faith community with meanings and components of a child's faith and suggests what to teach in early childhood. In addition, the developed scale can be a useful tool in objectively measuring a child's faith and proving the effectiveness of faith-related programs. Specifically, it is expected that researchers studying a spirituality scale development in other cultures will be able to conduct follow-up studies with modifications that reflect the content and processes of this study or their own cultural characteristics. Furthermore, in case of other religions, it would be possible to conduct future studies based on the meaning and factors of the spirituality/faith shown in this study.

**Table 1.** Spirituality/faith scales for children.

| Developer (Year), Scale Name | Factor, Item | Characteristic | Improvement | Follow-Up Study |
|---|---|---|---|---|
| Song (2005), Religious Development Interview Tool | - Good image<br>- Belief in Jesus and resurrection<br>- Heaven<br>- Prayer<br>- Forgiveness<br>-28 questions | - Open-ended questions based on Fowler (1981) and Goldman (1984)<br>- No response/don't know 0 point; preoperational thinking 1 point; concrete operational thinking 2 points | - Logical and theological soundness<br>- Subjectivity of scoring criteria | - Oh and Jung (2009)<br>- Lee and Kim (2010) |
| Howden (1992), Spirituality Assessment Scale | - Energy in life,<br>- Relationships with others and objects<br>- Integrated energy<br>- Positive perspective on the future<br>-28 items | - 6-point Likert scale<br>- 40–60 years olds | - Use for young children by translating and modifying the scale for agesd 40 to 60<br>-No questions about faith or God<br>-Inappropriateness of the factor-question | - Oh and Kang (2000)<br>- Lee (2002)<br>- Ko (2006)<br>- Park (2013)<br>- Kim (2014) |
| Fisher (2004), Feeling Good Living Life | - Self-concern<br>- Family<br>- Environment<br>- Relationship with God<br>- 4 items for each factor | - Ssubjects: 2018 children aged 5–12<br>- 5-point Likert scale | - 75% of the questions are about family, environment, self-emotions | - Lee and Lee (2016) |
| Stoyles et al. (2012), The Spiritual Sensitivity Scale for Children | - Outward focus<br>- Inward reflective focus<br>-23 items | - 4-point Likert scale | - No question related to faith or God. | |
| Moore et al. (2016), Children's Spirituality Measure | - Comfort<br>- Omnipresence<br>- Duality<br>- 27 items | - Subject: Children from various religious backgrounds<br>- Exploratory factor analysis only<br>- Measured by parents | | |
| Tratner et al. (2017a), Childhood Religious Experience Inventory | - Assurance<br>- Disapproval and punishment<br>- Social involvement<br>- Encouraged skepticism<br>- 16 items | - Subject: 18–34 years old | - Measuring parental influence, not child's spirituality<br>- Adult's recalling as a child<br>- Exploratory factor analysis only | |

Based on this research objective, the research questions are:

1.  What are the factors of a faith scale for young children?
2.  What is the validity of the faith scale for young children?
3.  What is the reliability of the faith scale for young children?

## 2. Results

### 2.1. Data Screening

To assess normality, the distribution of the data was examined with respect to skewness and kurtosis. Skewness ranged from $-2.660$ to $-0.372$, and kurtosis ranged from $-0.814$ to $8.611$. In case the absolute value is less than three for skewness and the absolute value is less than 10 for kurtosis, it is not considered to be an extreme case (Bae 2017). Normality, therefore, is not too badly violated.

### 2.2. Exploratory Factor Analysis

Exploratory factor analysis was conducted with 40 items obtained from a content validity test and pilot study to extract the new factor structure and to examine the construct validity. Initially, the factorability of the items was examined by checking the Kaiser–Meyer–Olkin Measure of Sampling (KMO) value (0.953) and Bartlett's test value ($\chi^2(300) = 6943.271$, $p = 0.000$). These results mean that factor analysis is appropriate because the KMO value is closer to 1 and Barrett's spherical verification result is statistically significant (Hwang et al. 2017). Items with factor loadings below 0.40 or items having high factor loadings more than one factor were removed from the analysis. Questions deleted in this process include: (5) I obey God's word; (8) I listen to my parents and teachers and obey them; (10) God gives me what I really need; (12) I happily give to God's ministries; (13) I pray for other people (such as friends and family); (15) God wants me to help and serve others; (18) I take care of the natural world created by God (not picking flowers, not bothering animals, not throwing littering); (21) I like Sunday school teachers and pastors; (26) I pray and read the Bible every day; (29) I worship with the right attitude and posture; (31) I love and make my parents happy; (32) I help when I see a difficulty; (35) I can say "No!" to wrong; (37) I give snacks or toys to my friends; (40) I will believe in God even if I become a grandfather or grandmother.

A principal components analysis using Varimax rotation was performed. A total of three factors were extracted and rotated, and the cumulative variance explained was 59.395. The first factor, *confessional faith life*, included sixteen items, representing 33.038%. The second factor, *missional life*, included five items, representing 13.330%. The third factor, *distinctive life*, included four items, representing 13.028%. Rotated factor loadings, variances, and eigenvalues are provided in Table 2.

### 2.3. Confirmatory Factor Analysis

2.3.1. Model Fit

An initial confirmatory factor analysis was conducted to assess 25 items, under three factors, based on exploratory factor analysis results. However, the original model fit failed to meet the standards ($\chi^2(272) = 1030.04$, $p = 0.000$); standardized root-mean square residual (SRMR) = 0.049; goodness-of-fit index (GFI) = 0.839; comparative fit index (CFI) = 0.888; Turker–Lewis index (TLI) = 0.877; root mean square error of approximation (RMSEA) = 0.080 (low 0.075; high 0.085). By examining modification indices, the study found that some covariances of errors are high; e20 and e21, e17 and e18, e12 and e16, e7 and e16, e7 and e12, and e4 and e5 are 58.810, 34.478, 26.571, 33.396, 37.045, and 28.627, respectively. After reviewing the similarities in the meaning of the corresponding questions, those errors were correlated. With these changes, the model produced a reasonably good fit: $\chi^2(266) = 814.166$, $p = 0.000$; SRMR = 0.042; GFI = 0.870; CFI = 0.919; TLI = 0.909; RMSEA = 0.069 (low 0.064; high 0.074).

**Table 2.** Factor loadings.

| Items | Factor 1 | Factor 2 | Factor 3 | Communality |
|---|---|---|---|---|
| Factor 1: *Confessional Faith Life* | | | | |
| 2. I feel that God loves me. | 0.762 | 0.253 | -0.034 | 0.646 |
| 24. God is precious to me. | 0.753 | 0.222 | 0.249 | 0.678 |
| 23. I believe that God is alive. | 0.753 | 0.246 | 0.229 | 0.674 |
| 34. I am glad I believe in Jesus. | 0.745 | 0.236 | 0.341 | 0.726 |
| 1. God knows my thoughts and feelings. | 0.739 | 0.173 | 0.184 | 0.611 |
| 33. I know that the Bible is the word of God. | 0.738 | 0.155 | 0.366 | 0.703 |
| 9. I believe that Jesus died on the cross to forgive my sins. | 0.708 | 0.230 | 0.256 | 0.620 |
| 19. I, who believe in Jesus, will go to heaven. | 0.689 | 0.278 | 0.116 | 0.566 |
| 36. God wants everyone to know and worship Him. | 0.677 | 0.135 | 0.333 | 0.588 |
| 3. I want to be a proud child to God. | 0.665 | 0.125 | 0.222 | 0.508 |
| 14. I think that God is the one who gives sunshine and rain. | 0.652 | 0.162 | 0.331 | 0.561 |
| 27. I am a precious person, created by God. | 0.642 | 0.135 | 0.347 | 0.551 |
| 25. God created me through my parents. | 0.641 | 0.300 | 0.298 | 0.589 |
| 17. I am grateful to see the world God has created. | 0.629 | 0.232 | 0.372 | 0.588 |
| 16. I am happy when I think about God. | 0.574 | 0.300 | 0.372 | 0.557 |
| 4. God helps me when I feel distressed or sad. | 0.544 | 0.321 | 0.284 | 0.480 |
| Factor 2: *Missional Life* | | | | |
| 7. I can tell anyone in the kindergarten or childcare center I believe in God. | 0.085 | 0.824 | 0.097 | 0.695 |
| 22. I tell my friends about Jesus. | 0.166 | 0.747 | 0.175 | 0.616 |
| 30. I think and talk about God. | 0.332 | 0.663 | 0.234 | 0.604 |
| 6. I like going to church and I wait for it. | 0.353 | 0.547 | 0.158 | 0.448 |
| 39. I like to worship. | 0.398 | 0.477 | 0.337 | 0.449 |
| Factor 3: *Distinctive Life* | | | | |
| 28. I feel sorry toward God when I do something bad. | 0.351 | 0.052 | 0.742 | 0.677 |
| 11. I ask God for forgiveness when I do something wrong. | 0.244 | 0.229 | 0.742 | 0.662 |
| 20. I, a child of God, do not lie. | 0.201 | 0.318 | 0.563 | 0.458 |
| 38. I pray to God when I have difficulties. | 0.329 | 0.382 | 0.540 | 0.546 |
| % of Variance | 33.038 | 13.330 | 13.028 | |
| Eigenvalues | 12.214 | 1.592 | 1.043 | |

With the final model, the standard coefficients of each factor appeared as 0.65–0.85, 0.54–0.75, and 0.57–0.72 for factors 1, 2, and 3, respectively. The standard coefficients and graphic illustrations of the three-factor model with 25 items are presented in Figure 1.

2.3.2. Validity Test

Confirmatory factor analysis was used to assess the model with respect to construct validity, including convergent and discriminant validity (Cho 2016a). Convergent and discriminant validity were conducted using both an examination of observed correlations and confirmatory factor analysis. In regard to convergent validity, the averaged variance extracted (AVE) and construct reliability (CR) scores need to be higher than the cutoffs of 0.5 and 0.7, respectively (Cho 2016a; Yu 2012). Table 3 provides evidence of convergent validity.

Factor 1 and Factor 2 ($r = 0.714$, $p = 0.01$), Factor 1 and Factor 3 ($r = 0.638$, $p = 0.01$), and Factor 2 and Factor 3 ($r = 0.606$, $p = 0.01$) showed a significant correlation. For discriminant validity, squared correlation coefficients should be less than AVE (Yu 2012). The results of this study satisfy discriminant validity as shown in Table 4.

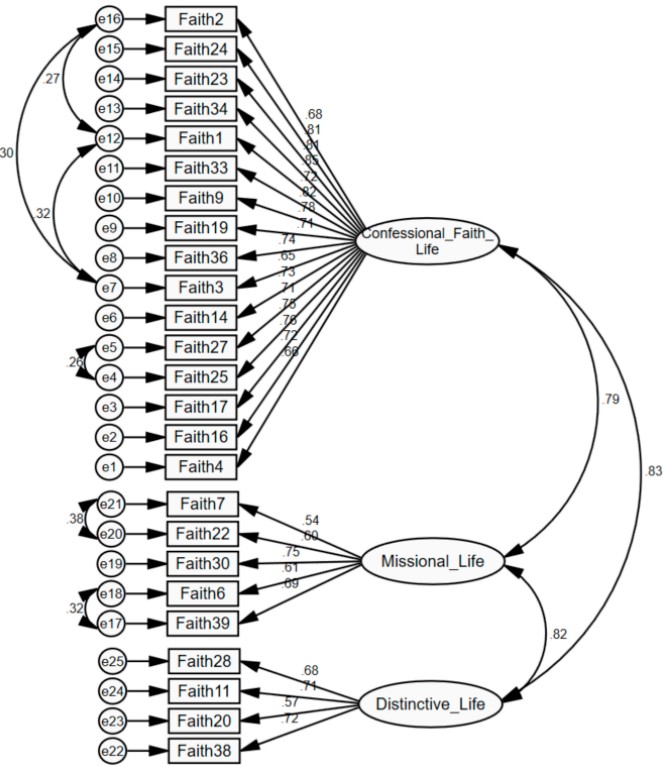

**Figure 1.** The path diagram of model.

**Table 3.** Convergent validity.

| Factor | Item | Standardized Regression Weight | t-Value | CR (Construct Reliability) | AVE (Averaged Variance Extracted) |
|---|---|---|---|---|---|
| Confessional Faith Life | Faith 4 | 0.657 | Fix | | |
| | Faith 6 | 0.72 | 13.556 | | |
| | Faith 17 | 0.757 | 14.139 | | |
| | Faith 25 | 0.754 | 14.102 | | |
| | Faith 27 | 0.711 | 13.413 | | |
| | Faith 14 | 0.731 | 13.732 | | |
| | Faith 3 | 0.646 | 12.328 | | |
| | Faith 36 | 0.74 | 13.882 | 0.993 | 0.893 |
| | Faith 19 | 0.713 | 13.442 | | |
| | Faith 9 | 0.78 | 14.513 | | |
| | Faith 33 | 0.825 | 15.188 | | |
| | Faith 1 | 0.72 | 13.554 | | |
| | Faith 34 | 0.849 | 15.549 | | |
| | Faith 23 | 0.813 | 15.018 | | |
| | Faith 24 | 0.807 | 14.915 | | |
| | Faith 2 | 0.676 | 12.831 | | |
| Missional Life | Faith 39 | 0.692 | Fix | | |
| | Faith 6 | 0.606 | 13.404 | | |
| | Faith 30 | 0.747 | 12.974 | 0.963 | 0.84 |
| | Faith 22 | 0.597 | 10.753 | | |
| | Faith 7 | 0.536 | 9.717 | | |
| Distinctive Life | Faith 38 | 0.716 | Fix | | |
| | Faith 20 | 0.575 | 10.891 | 0.971 | 0.894 |
| | Faith 11 | 0.715 | 13.37 | | |
| | Faith 28 | 0.68 | 12.778 | | |

$\chi^2(266) = 814.166$, $p = 0.000$; SRMR = 0.042; GFI = 0.870;
CFI = 0.919; TLI = 0.909; RMSEA = 0.069

**Table 4.** Discriminant validity.

| KERRYPNX | Factor 1 | Factor 2 | Factor 3 | AVE |
|---|---|---|---|---|
| Factor 1 | | | | 0.893 |
| Factor 2 | 0.714 ** | | | 0.840 |
| Factor 3 | 0.638 ** | 0.606 ** | | 0.894 |
| Total | 0.958 ** | 0.830 ** | 0.802 ** | |

** $p = 0.01$.

### 2.4. Reliability Test

The Cronbach's alpha coefficients were calculated to verify the reliability of the final three factors, including the 25 questions. Factor 1 is 0.951, factor 2 is 0.766, factor 3 is 0.795, and the overall value is 0.949 as shown in Table 5.

**Table 5.** Reliability.

| | Items | *M* (*SD*) | Cronbach's $\alpha$ |
|---|---|---|---|
| Factor 1 | 1–16 | 4.56 (0.58) | 0.95 |
| Factor 2 | 17–21 | 4.04 (0.79) | 0.77 |
| Factor 3 | 22–25 | 4.06 (0.80) | 0.80 |
| Total | 1–15 | 4.38 (0.59) | 0.95 |

### 2.5. Final Version

The measurement tool developed in this study is to measure preschool child's Christian faith and is named *The Faith Scale for Young Children* (*FSYC*). Young children's faith is defined as a system of life that manifests itself in the attitudes and actions of knowing, believing in, and loving, God. The final *Faith Scale for Young Children* derived through the research procedures consists of three factors and 25 questions as shown in Table 6.

**Table 6.** Final factor and item.

| Factor | Item |
|---|---|
| Confessional Faith Life | 1. I feel that God loves me.<br>2. God is precious to me.<br>3. I believe that God is alive.<br>4. I am glad I believe in Jesus.<br>5. God knows my thoughts and feelings.<br>6. I know that the Bible is the word of God.<br>7. I believe that Jesus died on the cross to forgive my sins.<br>8. I, who believe in Jesus, will go to heaven.<br>9. God wants everyone to know and worship Him.<br>10. I want to be a proud child to God.<br>11. I think that God is the one who gives sunshine and rain.<br>12. I am a precious person, created by God.<br>13. God created me through my parents.<br>14. I am grateful to see the world God has created.<br>15. I am happy when I think about God.<br>16. God helps me when I feel distressed or sad. |
| Missional Life | 17. I can tell anyone in the kindergarten or childcare center I believe in God.<br>18. I tell my friends about Jesus.<br>19. I think and talk about God.<br>20. I like going to church and I wait for it.<br>21. I like to worship. |
| Distinctive Life | 22. I feel sorry toward God when I do something bad.<br>23. I ask God for forgiveness when I do something wrong.<br>24. I, a child of God, do not lie.<br>25. I pray to God when I have difficulties. |

## 3. Discussion

In this study, *the Faith Scale for Young Children* was developed and validated, and three factors were derived; *confessional faith life (knowing God, responding to God)*, *missional life,* and *distinctive life* (hereafter referred to as Factors 1, 2, and 3, respectively). Factor 1 consists of 16 questions, including "Jesus died on the cross to forgive my sins." and "I am a precious person, created by God." and includes two concepts, knowing God's work and a child's response to Him. Factor 2 consists of five questions, including "I like going to church and I wait for it." and "I tell my friends about Jesus." while Factor 3 consists of four questions, including "I ask God for forgiveness when I do something wrong." and "I, a child of God, do not lie." These results are discussed below by comparing them with previous research or with the arguments of scholars.

The original foundation of this scale was based on knowing ("who God is" and "what God has done"), loving ("You can have a relationship with God."), and living ("You can be all God wants you to be." and "You can do all God wants you to do.") presented by Trent et al. (2000). The results reflect three of the components presented by Trent et al. (2000) by integrating knowing and loving into Factor 1 and living into Factors 2 and 3. Trent et al. (2000) distinction and the results of this study are consistent with Boa (2020) views that loving God completely involves intellect (know Thee more clearly), emotion (love Thee more dearly), and will (follow Thee more nearly). It is also in line with the view of Sifers et al. (2012), who saw that spirituality is "a set of believes, practices, guidelines for well-being, and a sense of experience." (p. 209)

Factor 1, *confessional faith life*, includes knowing what God has done and a child's response to Him. The result that knowing God's work included in this faith scale is supported by the views of leading researchers (Beers 1986; Lee and Lee 1988; Trent et al. 2000) who presented biblical knowledge and theological concepts that young children should know. It is also in line with the opinion of Jang (2002), who asserted that the true faith starts from knowing God from an early age and living for His glory. Factor 1 also includes questions confirming the young children's response to God. Affective domain, which reflects the feelings and will of young children, has been emphasized as an important part of faith (Kim 2008; Trent et al. 2000; Yang 2008). According to Joo (2019), the emotions that children feel as they imagine the worship atmosphere and Bible stories are internalized and the basis of spirituality, and this affective map acts as a motivation to interpret certain situations, judge, and execute desirable behaviors. Yang (2013), therefore, emphasizes the concept of formation rather than teaching in spiritual education during childhood. While teaching is an act that helps learners acquire certain knowledge, skills, or attitudes, forming is a process that prepares and forms body, emotions, and cognition to the content of faith before teaching. Given that early childhood is a time when basic images and affective maps of life are formed, children form a basic attitude toward others, the world, and God through interaction with significant others, and they likewise form a basic system of spirituality. Children's faith can then grow into a holistic spirituality when they learn theological concepts and biblical values that are consistent with the emotional map they have already embodied (Joo 2019).

Factor 2, *missional life*, includes questions about worship and evangelism to God. Worship is one of the most important elements of faith (Jang 2011; Joo 2019; Kim 2020a; Yang 2008). This argument about worship relates to the identity of believers. Born again Christians want to worship God and aspire to be holy because they have a new character to find God (Park 2017). Believers are temples and a temple is first of all a worship place (1 Cor. 3:16–17; 6:19–20; Eph. 2:19–22; Averbeck 2008). The benefits of worship are also linked to faith. God rejoices in the worship of the Christians and shows his presence and companionship in the meeting. Through the grace of God's presence, Christians' faith becomes solid and their lives change (Park 2017). Children learn what God is like through worship (Jang 2011). The evangelism of the gospel has been mentioned by several scholars as a theological concept to be known in early childhood. Previous researchers and Lifeway emphasized the importance of evangelism as follows, although the expressions are slightly

different: "God wants me to tell others about Jesus." (Beers 1986); "God loves everyone and wants everyone to love God." and "God wants the child to tell others about Jesus." (Lee and Lee 1988); "I can tell people in my community that Jesus loves them." (LifeWay n.d., https://www.lifeway.com/en/product/levels-of-biblical-learning-poster-birth-high-school-pkg-10-P005817005, accessed on 18 February 2021)

Factor 3, *distinctive life*, consists of questions about the sensitivity of sin and the need to pray to God when one is in trouble. The finding is supported by Self (1986), who claimed that 4- and 5-year-old children distinguish between right and wrong. This finding is also in line with Kim (2020a) finding where 17 Christian education experts were interviewed and stated that the discernment of good and evil is one of the theological concepts understood by young children.

Finally, the researcher would like to finish this paper by presenting the spiritual characteristics of young children proposed by Wilson (1991). According to him, 4-year-old children form the concept of right and wrong, learn that wrong behavior is a sin in God's sight, and feel sorry about their sin. In addition, young children understand God literally, realize God's greatness and goodness through worship, and pray for what happens in life. The spiritual features he mentioned and the contents of the faith scale developed in this study are largely consistent.

The subjects and developing process of this scale were planned for the Sunday school teachers, ministers, or teachers at Christian kindergartens to assess the faith of children whom they teach. However, due to the COVID-19 pandemic, it was difficult for young children to attend church and kindergarten, making this method impossible. Therefore, in this study, the scale was developed through parental assessment. In future studies, the scale that can be evaluated more objectively by teachers or ministers needs to be developed.

## 4. Materials and Methods

### 4.1. Research Subject

The subjects of this study were 434 young children who are attending church but have not begun elementary school. The average age of young children was 67.41 (*SD* = 10.26) month, with 186 boys (42.9%) and 248 girls (57.1%). Children's faith level corresponding to each question was rated by their parents using a 5-point Likert scale. Forty-two fathers (9.7%) and 392 mothers (90.3%) were evaluated for their child's faith.

### 4.2. Research Precedures

#### 4.2.1. Factor and Item Generation

Opinions on the contents of early childhood faith are somewhat different among scholars, but they tend to involve opinions on knowledge, attitude, and behavior related to theological concepts. This researcher identified key factors and the corresponding preliminary items by referring to early childhood faith contents presented in existing studies. This study builds upon the idea of Trent et al. (2000) who used *knowing, loving, and living* as a main frame of reference. The contents of early childhood faith presented by other scholars are as follows: Beers (1986)—God, Jesus, the Bible, home and parents, church and Sunday school, others, angles and last things; Fisher (2004)—self, others, environment, god; Kwon (2013)—God, Jesus, Holy Spirit, prayer, worship, creation, faith and spiritual life; Lee and Lee (1988)—God, Jesus, Holy Spirit, church, relationship with others, sins and forgiveness, resurrection, eternal life, Bible, worship; Lifeway—God, Jesus, Bible, creation, family, self, church, community and world; Moore et al. (2016)—comfort, omnipresence, duality; Tratner et al. (2017a)—assurance, disapproval and punishment, social involvement, encouraged skepticism. Based on these studies, 65 preliminary items were developed, with 23 questions for *knowing*, 22 questions for *loving*, and 20 questions for *living*. The main materials referenced in the construction of the preliminary factors and items of this study are shown in Figure 2.

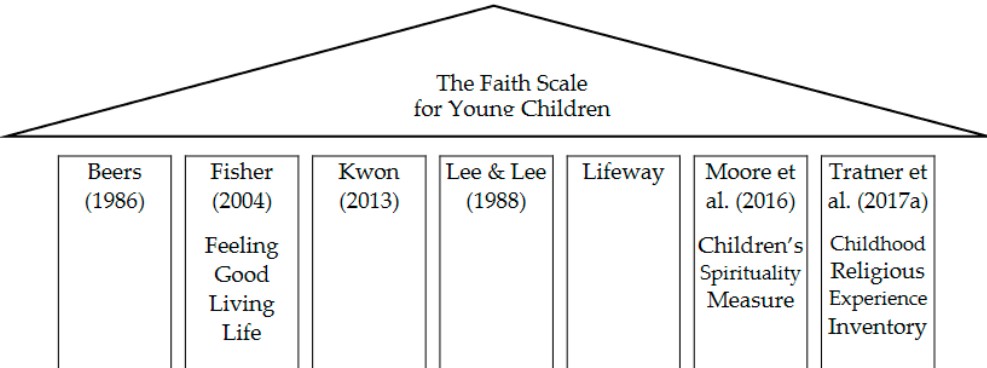

**Figure 2.** Theoretical foundation.

### 4.2.2. Content Validity Test

Sixty-five items, developed by referring to existing theories and based on the knowledge and experience of the researcher, were evaluated by seven professionals with doctoral degrees in education. Four of the panelists majored in Christian Education and are church ministers, two majored in early childhood education and serve as directors of Christian kindergartens, and one panelist is a missionary with a doctoral degree in education. They evaluated the preliminary items using a 5-point Likert scale and were asked to add additional questions that this researcher omitted. In line with this process, 40 questions were selected, including the additional questions from experts such as "God helps me when I feel distressed or sad." and "I give snacks or toys to my friends."

### 4.2.3. Pilot Study

It is recommended that 30 or more samples are needed for a pilot study (Browne 1995). Hence, the pilot test was conducted involving 45 children through Google surveys. The pilot test revealed one question that was unclear or difficult to rate. It was revised accordingly: "I can say "No!" to something I shouldn't do by thinking of God." to "I can say "No!" to wrong."

### 4.2.4. Main Study

The main study was conducted using both Google surveys and paper-based surveys. Four hundred thirty-four copies of the survey were retrieved by forwarding the link of online surveys or printed paper surveys with tokens to parents.

### 4.2.5. Data Analysis

To analyze the gathered data, SPSS 22 was used for descriptive statistics and exploratory data analysis. AMOS 22 was also used for confirmatory data analysis.

## 5. Conclusions

This study focused on the development and validation of the *Faith Scale for Young Children*. The research procedures and results are presented according to research questions. The factors of the *Faith Scale for Young Children* according to the research question 1 are *confessional faith life*, *missional life*, and *distinctive life*. The validity and reliability of this scale, corresponding to research question 2 and 3 have been proven by presenting the required results and numerical values. Yang (2008) suggested that Christian education for young children should move toward education in which they meet God, worship Him, talk to Him, and relate to Him; that is, education in which God is experienced. Against this backdrop, *the Faith Scale for Young Children* was developed, which is based on knowing God, responding in a positive manner, and living on the right standard in front of Him. Based on the results, this study proposes to know, believe, and serve God in the direction of Christian education for early childhood.

**Funding:** This work was supported by the Ministry of Education of the Republic of Korea and the National Research Foundation of Korea (NRF-2019S1A5A8037639).

**Institutional Review Board Statement:** Ethical review and approval were waived for this study because the ethics committee of the school to which this researcher belongs did not function. In my country, this regulation was not mandatory when publishing academic journals, so I failed to comply with it. But I swear I didn't do anything against the regulations.

**Informed Consent Statement:** Informed consent was obtained from all subjects involved in the study.

**Data Availability Statement:** Not applicable.

**Conflicts of Interest:** The author declares no conflict of interest.

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
