# Peer review of "Development and Validation of a Faith Scale for Young Children"

_religions, doi:10.3390/rel12030197_

Round 1
Reviewer 1 Report
Thank you for this interesting piece. I am not a statistician so I will not pass comment on the data handlings etc. Re the general approach and conceptual understanding, I have the following suggestions:
- Consider briefly in the Introduction how far the survey is applicable to other religious traditions (or not, as the case may be.)
- Perhaps include a short section in the Introduction which explores the reason why such data is helpful to a faith community.
- p. 1, final line and rest of the paragraph on p. 2: Are ‘faith’ and ‘spirituality’ the same thing? This is an area which would benefit from some expansion. Perhaps add a short section which recognises this.
- p.4 para’ 1 ‘Research questions’: There are three important questions but it would be important to present the responses to the questions in the Conclusions. Check spelling: ‘Qustions’ should be ‘questions’.) Add a section to the Conclusion (p. 11) which presents the answers, even if they are not conclusive.
- p. 9, end of para’ 1: Consider if teaching is part of formation instead of being simply a preparation for it, as is implied. The place of worship in teaching/learning in religious matters needs careful and nuanced handling. There is no need to go into this in too much depth but it would be good to show some understanding, with appropriate references, to the field.
Many thanks.
Author Response
Based on your evaluation, I did my best to modify the paper.

Reviewer 2 Report
This is a very good quantitative study. I did not find any major areas that are in need of improvement.
The discussion of the data is clear and concise and easy to follow - there are some formatting issues where the text seems off sides from alignment, but overall, the graphs are presented well.
If there was one area for improvement it would be the conclusion - perhaps a further discussion or call for one on what a practitioner/educator could do with this information would be good. This especially would encourage the conversation in relation to the findings to continue. It also would be good to see how these findings are predicted to relate to other children out of other cultures.
Author Response

(The authors gave the same response as above.)

Reviewer 3 Report
This artical has a huge sweep in the number of studies it covers, but
it covers them in such a quick glance it's hard for someone like me to
get a hold of what the author is getting from them. I'd like a
narrative summary explaining this and a clearer statement of, having
done this review of the studies, how the author believes a child's
faith is developed, and under what circumstances it's shaped.
I do not have enough statistical knowledge to be able to judge well and would have like clearer narrative explanation of where this research led author.
Author Response
|
Thank you for the suggestion for this study. This study, however, focuses on what a child’s faith consists of, and this researcher predicts that mentioning how and what circumstances will deviate from focus and increase volume significantly. The researcher's recent studies (Kim 2020a; Kim 2020b) explain influential factors well (Family-parents, grandparents; church-ministers and teachers; weekday educational institution; friends, etc.), so I think the readers can get help if they refer to the data. |

Round 2
Reviewer 1 Report
I have below highlighted the original request and my assessment of how the response has addressed it.
Request 1
Consider briefly in the Introduction how far the survey is applicable to other religious traditions
Response 1
While some mention has been made of general applicability on p. 10, the Introduction needs to mention this too
Request 2
Perhaps include a short section in the Introduction which explores the reason why such data is helpful to a faith community.
Response 2
Satisfactory response
Request 3
- 1, final line and rest of the paragraph on p. 2: Are ‘faith’ and ‘spirituality’ the same thing? This is an area which would benefit from some expansion. Perhaps add a short section which recognises this.
Response 3
A footnote has been added but there needs to be something deeper. Perhaps another half page of so, setting out the arguments, not just references.
Request 4
p.4 para’ 1 ‘Research questions’: There are three important questions but it would be important to present the responses to the questions in the Conclusions. Check spelling: ‘Qustions’ should be ‘questions’.) Add a section to the Conclusion (p. 11) which presents the answers, even if they are not conclusive.
Response 4
Number the responses 1. 2 and 3 to match the initial questions.
Request 5
â–ª p. 9, end of para’ 1: Consider if teaching is part of formation instead of being simply a preparation for it, as is implied. The place of worship in teaching/learning in religious matters needs careful and nuanced handling. There is no need to go into this in too much depth but it would be good to show some understanding, with appropriate references, to the field.
Response 5
There is the beginning of a response here. More development is needed: the references mentioned are not helpful as they stand but need to be explained.
For example:
‘Spiritual understandings and practices are more caught than found.’ -what does this actually mean? It seems that this needs to be interrogated as a principle which reference to wider educational literature.
‘Worship is one of the most important elements of faith (Jang 2011; Joo 2019; Kim 2020a; Yang 2008).’ What does this actually mean? Why is it ‘true’? Could there be another angle to be considered?
The changes I suggest above require some wider reading and might take a few weeks. This will give the author(s) time to explore the wider literature. There is no need to rush as this is an important topic.
presents the answers, even if they are not conclusive.
Author Response
|
Modification Request |
Revision |
Line Number |
|
While some mention has been made of general applicability on p. 10, the Introduction needs to mention this too. |
I moved the contents on page 10 to introduction, and rearranged them in a logical order after combining with the significance of this study previously recorded.
|
131~ |
|
Perhaps include a short section in the Introduction which explores the reason why such data is helpful to a faith community.
|
Satisfactory response
|
|
|
A footnote has been added but there needs to be something deeper. Perhaps another half page of so, setting out the arguments, not just references. |
Although the meaning of spirituality varies depending on the individual and religions, the universal meaning of spirituality is the process of pursuing the ultimate value an individual believes in and creating a mature life by integrating this value in the real life (You 2009). Citing Toon (1989), Anthony (2006) defines spirituality as “all about the human search for identity, meaning, purpose, God, self-transcendence, mystical experience, integration and inner harmony.” (p. 6) Yeon (2004) defined spirituality as an insight into the transcendent aspects of human beings and their relationships with all worlds. Therefore, spirituality can be associated with God, nature, others, and the surroundings (Paul-Victor and Treschuk 2020). According to Lunn (2009) who distinguished between spirituality and faith, spirituality is an individual's beliefs and experiences in the supernatural realm, while faith is the belief or trust in a transcendent reality. Faith is an act of spiritual knowledge that involves beliefs with strong intellectual tendencies, formed by influential spiritual mediators or indirect religious and cultural factors (Kim 2006). According to Paul-Victor and Treschuk (2020), faith is often associated with spirituality and religion, but it is more personal, subjective, and deeper than these. As such, some scholars distinguish between spirituality and faith and some preferred terms. However, this study views faith and spirituality as similar and compatible terms based on Christian truth and tradition.
|
109~ |
|
Number the responses 1. 2 and 3 to match the initial questions.
|
The factors of the Faith Scale for Young Children according to the research question 1 are confessional faith life, missional life, and distinctive life. The validity and reliability of this scale corresponding to research question 2 and 3 have been proven by presenting the required results and numerical values. |
377~ |
|
There is the beginning of a response here. More development is needed: the references mentioned are not helpful as they stand but need to be explained.
|
This argument about worship relates to the identity of believers. Born again Christians want to worship God and aspire to be holy because they have a new character to find God (Park 2017). Believers are temples and a temple is first of all a worship place (1 Cor. 3:16-17; 6:19-20; Eph. 2:19-22; Averbeck, 2008). The benefits of worship are also linked to faith. God rejoices in the worship of the Christians and shows his presence and companionship in the meeting. Through the grace of God's presence, Christians' faith becomes solid and their lives change (Park 2017). |
270-275; 278~284 |
